# An Improved Algorithm for Measuring Nitrate Concentrations in Seawater Based on Deep-Ultraviolet Spectrophotometry: A Case Study of the Aoshan Bay Seawater and Western Pacific Seawater

**DOI:** 10.3390/s21030965

**Published:** 2021-02-01

**Authors:** Xingyue Zhu, Kaixiong Yu, Xiaofan Zhu, Juan Su, Chi Wu

**Affiliations:** 1Shandong Provincial Center for In-Situ Marine Sensors, Institute of Marine Science and Technology, Shandong University, Qingdao 266237, China; zhuxingyue@sdu.edu.cn (X.Z.); 201916201@mail.sdu.edu.cn (K.Y.); 230208487@seu.edu.cn (X.Z.); sujuan@sdu.edu.cn (J.S.); 2Aixsensor Co., 587 Jinghua Road, Dezhou 253500, China; 3Qingdao Institute of Marine Sensors, 1 Keji Road, Jimo, Qingdao 266237, China

**Keywords:** nitrate concentration, deep-UV optical sensor system, seawater monitoring, kernel partial least squares

## Abstract

Nowadays, it is still a challenge for commercial nitrate sensors to meet the requirement of high accuracy in a complex water. Based on deep-ultraviolet spectral analysis and a regression algorithm, a different measuring method for obtaining the concentration of nitrate in seawater is proposed in this paper. The system consists of a deuterium lamp, an optical fiber splitter module, a reflection probe, temperature and salinity sensors, and a deep-ultraviolet spectrometer. The regression model based on weighted average kernel partial least squares (WA-KPLS) algorithm together with corrections for temperature and salinity (TSC) is established. After that, the seawater samples from Western Pacific and Aoshan Bay in Qingdao, China with the addition of various nitrate concentrations are studied to verify the reliability and accuracy of the method. The results show that the TSC-WA-KPLS algorithm shows the best results when compared against the multiple linear regression (MLR) and ISUS (in situ ultraviolet spectrophotometer) algorithms in the temperatures range of 4–25 °C, with RMSEP of 0.67 µmol/L for Aoshan Bay seawater and 1.08 µmol/L for Western Pacific seawater. The method proposed in this paper is suitable for measuring the nitrate concentration in seawater with higher accuracy, which could find application in the development of in-situ and real-time nitrate sensors.

## 1. Introduction

With the development of the economy, agricultural fertilizers and various sewages are continuously discharged to the coastal seawater [1,2] and have led to the abnormal reproduction of algae and even red tide outbreaks. This abnormality does not only bring serious consequences to nature and human beings, but also results in huge losses to the local economy [3,4]. Eutrophication is one of the key factors that lead to red tide outbreaks, while the nitrate concentration is one of the most important parameters to characterize the eutrophication [5]. Therefore, developing nitrate sensors with high accuracy in a complex water is of great significance for the exploration of the mechanisms behind red tide outbreaks. However, it is still a challenge today for commercial in-situ nitrate sensors to meet the demand.

Scientists began to study nitrate in seawater in the early 1920s using laboratory chemical methods, which were time-consuming [6,7]. In the 1960s, researchers realized that the ultraviolet absorbance of seawater could be used as an important indicator to evaluate the quality of seawater [8]. Various technologies developed in the past decades can be classified into the following categories: field sampling plus laboratory testing technique, field testing with wet chemical analysis technique and field testing with in situ deep ultraviolet sensor analysis technology.

For the field sampling plus laboratory testing technique, the samples need to be brought to the lab and measured by the methods based on various chemical reagents or by analyzers, which include the spectrophotometry method [9,10,11,12], the liquid chromatography method [13,14,15], the fluorescence method [16,17,18] and the isotope detection method [19]. These laboratory techniques are highly accurate, but the samples can easily be contaminated in the process of collection and transportation.

To eliminate contamination during transportation, scientists in the 1960s developed an instrument based on the wet chemical analysis method to monitor and analyze seawater. By loading a pump on board, the seawater samples were pumped into the instrument for detection [20]. This method enabled the samples to be tested in an airtight environment, which laid the foundation for the automation and integration of the wet chemical analyzer [21]. Based on this, Ŕužička and Hansen first proposed the flow injection analysis (FIA) method to avoid the effect of bubbles when pumping seawater on board [22]. This approach has the advantage of being faster in measurement time comparing to prior arts, and greatly promoted the development of automatic on-line monitoring instruments [23]. However, the temperature and pressure of the samples change during the process of pumping, which may change physical properties of the seawater compositions. To solve this problem, scientists then developed the underwater in-situ analysis instruments, which can be applied to different measurement platforms [24,25]. In recent years, technology based on microfluidic chip was developed rapidly in the water quality detection field. This technology integrates the sample preparation, reaction, separation, detection, and other basic operation units of biological and chemical processes into a micro-scale chip, and automatically completes the whole process of analysis. This is suitable for the trace detection of nitrate in seawater, and is advantageous due to its low power consumption, high sensitivity, and fast speed [26,27,28,29]. However, the working time of wet chemical analysis depends on the stability and consumption rate of chemical reagents and it is hard to realize long-term (more than three months) in-situ monitoring. With the consideration of its high maintenance cost, its application for in-situ monitoring of marine environment is limited. 

In recent years, the ultraviolet absorption spectroscopy-based technique was developed in measuring nitrate in seawater, with the advantages of no chemical reagents, no pollution to environment, and ease of operation [30,31]. In 1998, Finch and his group developed the first ultraviolet spectrometer system capable of in-situ measurements, with the detection limit lower than 0.21 µM of nitrate in seawater [32]. After that, the SUV-6 in-situ nitrate sensor was established by optimizing the hardware and software of the system. In 2002, Johnson and his colleagues developed an ISUS (in situ ultraviolet spectrophotometer) system that was commercialized by Sea-Bird Scientific [33]. Some research institutions have also improved the nitrate in-situ sensor and conducted sea trial, which contributed to the improvement of measurement accuracy [34,35,36]. Currently, more and more in-situ nitrate monitoring instruments based on ultraviolet spectroscopy have been used in rivers and seawater, such as TriOS OPUS multi-parameter analyzer and SUNA sensor [37,38]. However, some research results showed that the actual measurement errors of these commercial nitrate sensors are higher than the parameters given in their specifications, for example, the SUNA and SUNA V2 combined data produced a 0.66 mg-N/L average absolute difference from 40 discrete samples collected from three deployment sites, which resulted in inaccurate results [39].

To overcome the above challenges, we propose a deep-ultraviolet spectroscopy-based nitrate measurement method with high accuracy. The nitrate concentration in the seawater samples from Western Pacific and Aoshan Bay, Qingdao of China were measured and predicted as a case study to verify the reliability and accuracy of the method. The measurement system is composed of a deuterium lamp, a signal acquisition module, and a signal processing module. By comparing the parameters from different models and prediction data, optimal algorithm for the system was selected. Such measurement is relatively simple and quick, as no chemical manipulation is required. The results show that the method proposed in this paper can improve the accuracy of the calculated concentrations when applied to input data within the ranges of the training input data but can possibly lead to runaway concentrations when applied to input data outside of the ranges of the training input data. The approach is reliable and applicable for nitrate determination, which lays a foundation for the development of marine environmental monitoring technology. 

## 2. Materials and Methods

### 2.1. Main Elements in Seawater

The composition of seawater is complex and contains many chemical substances. The main dissolved elements include Na^+^, Mg^2+^, Ca^2+^, K^+^, HPO_4_^2−^, SO_4_^2−^, Cl^−^, Br^−^, HCO_3_^−^, etc. Besides these dissolved elements, organic matter is also an important substance in seawater. It mainly comes from two parts. One is produced by the ocean itself, including metabolites, decomposition, and debris of marine organisms. The other is artificially introduced from land sources, including organic substances and organic pollutants produced by human life and production activities. Organic matter concentration in seawater also has the absorbance in the UV wavelength range, with the main component of humate [40,41,42,43]. Therefore, the ultraviolet absorption characteristics of some main elements, including sodium chloride, magnesium sulfate, sodium bicarbonate, sodium bromide, sodium nitrate, sodium humate, sodium dihydrogen phosphate, and sodium nitrite, have to be measured and analyzed.

### 2.2. Optical Measurement

#### 2.2.1. Measurement Principle

The model of ultraviolet spectrophotometry is established based on Lambert Beer’s law. The absorbance $A_{\lambda }$ of the sample at each wavelength λ is given by
(1)$$A_{\lambda }=-\log_{10}\left(\frac{I_{\lambda }-I_{D}}{I_{\lambda ,0}-I_{D}}\right),$$
where $I_{\lambda }$ is the light intensity of the sample measured by the detector at wavelength λ, $I_{\lambda ,0}$ is the light intensity of deionized water measured by the detector at wavelength λ, and $I_{D}$ is the dark current of the detector. The sample absorbance $A_{\lambda }$ can be expressed as the sum of the absorbance of each chemical element in sample according to the additivity of absorbance, that is
(2)$$A_{\lambda }=A_{1}+A_{2}+\ldots +A_{j}+\ldots +A_{M}=b\left(\sum_{j}\varepsilon_{\lambda ,j}C_{j}\right),$$
where $\varepsilon_{\lambda ,j}$ is the molar absorptivity of $j_{th}$ element in seawater, b is the optical path, and $C_{j}$ is the concentration of the $j_{th}$ element. However, the seawater elements are complex, and the absorption spectrum of each element ranges overlap, which cannot be effectively separated. Therefore, based on the above principles, the kernel partial least squares algorithm is used to establish the calculation model, which will be introduced in Section 2.3.2.

#### 2.2.2. Description of the Measurement System

The measurement system is shown in Figure 1. The light emitted by the deuterium lamp (DH-2000-DUV, Ocean optics, Orlando, FL, USA) is attenuated by the adjustable optical fiber attenuator (FVA-UV, Ocean optics, Orlando, FL, USA) and is further transmitted into the optical fiber splitter (customized, Wyoptics Co., Ltd., Shanghai, China). Then the signal is reflected by the reflection probe and returned to the fiber with the concentration information from the sample. Finally, the data are collected and calculated by the data processing module. Deionized water is in sample cell 1 and the measurement sample is in sample cell 2. Temperature and salinity sensors are placed in sample cell 2. The temperature and salinity sensors are utilized to measure and transmit the data to the data processing module. When measuring the data for establishing the calculation model, the temperature control module is needed to control the temperature change of samples uniformly. The UV spectrometer (QE Pro, Ocean Optics Inc., Orlando, FL, USA) is used here to collect the data. The specifications of the light source and the spectrometer are shown in Table 1.

The choices of hardware components are shown as follows.

Light source

The light source is selected based on wavelength range, luminous stability, output power, and source lifetime. Deuterium lamps generate continuously stable light output in the deep ultraviolet region for a long time. Therefore, the DH-2000-DUV light source was selected with an output wavelength range of 190–400 nm, a lifetime of 1000 h, and a light output stability of less than 5 × 10^−6^ peak to peak (0.1–10.0 Hz).

2.UV fiber splitter module

The UV fiber splitter module is one of the core structures of the system. The use of a reference optical path can not only reduce the system errors and detect fluctuations of the light source, but also simultaneously measure the light attenuation of deionized water, which improves the accuracy of the measured absorbance. The structure and size of the fiber splitter module shown in Figure 2 is made of quartz fiber with a core diameter of 400 µm. Port 4 is the input port of light source, port 3 and port 5 are the output ports of signal, and port 1 and port 2 are connected with underwater reflection probes, which are used to detect and return the signal from the sample and deionized water, respectively. The reflection probe is made of stainless steel, with one end of a mirror and the other end of a fiber port. The optical paths of the underwater reflection probes are set to 10 mm.

3.Temperature control module

To keep the temperature of the measured seawater sample stable, a customized small thermostat bath (Shanghai Pingyuan Scientific Instrument Co., Ltd., Shanghai, China) is used in the system. The temperature is controlled by the PID algorithm and auto tuning technology. In addition, the thermostat bath has an internal and external circulation system to ensure the constant temperature. The temperature range can be controlled from −5 °C to 100 °C with a precision of 0.05 °C.

4.Spectrometer

The spectrometer (QE Pro, Ocean Optics Inc., USA) is used with an optical resolution of 0.4 nm in the wavelength range from 200 nm to 385 nm. The QE Pro is a spectrometer with high sensitivity and low stray light, which is ideal for the absorbance application. It has good thermal stability, and functions of onboard spectral buffering, automatic integration, and multi-sampling, which could amplify weak signals and eliminate random noise. 

### 2.3. Model Establishment Method

#### 2.3.1. Temperature and Salinity Correction Algorithm

As the absorbance of seawater varies with the temperature, it is necessary to establish temperature and salinity correction (TSC) algorithms for the calculation model. The absorbance of low nutrient seawater (LNS, created by filtering seawater from Western Pacific using 0.45 μm filter membrane) samples are all normalized to psu (Practical Salinity Unit) = 35, which is
(3)$$A_{W,T}=A_{m}\times \frac{S_{S\tan dard}}{S_{m}},$$
where $S_{m}$ and $A_{m}$ are the measured salinity and absorbance of seawater samples respectively, and $S_{s\tan dard}$ equals 35. Based on the measured data and polynomial regression equation, the model of LNS can be established as
(4)$$A_{W,T}=p_{00}+p_{10}\times W+p_{01}\times T+p_{20}\times W^{2}+p_{11}\times W\times T+\ldots ,$$
where W and T are the centralized wavelength and temperature, $p_{00}$, $p_{10}$, $p_{01}$, $p_{20}$, $p_{11}$ are the regression parameters of the model. As the temperature dependence of LNS absorbance is dominated by the bromide absorbance, its absorbance trend is consistent with bromide [44], therefore the temperature influence is included in $A_{W,T}$.

To ensure the accuracy of the nitrate modeling data, it is necessary to first obtain the absorbance caused by temperature. Therefore, the molar absorptivity of sea salt is introduced and can be calculated as [37]
(5)$$E_{W,T_{m}}=\frac{E_{W,T_{cal}}\times A_{W,T_{m}}}{A_{W,T_{cal}}},$$
where $E_{W,T_{cal}}$ and $A_{W,T_{cal}}$ are the molar absorptivity and normalized absorbance of calibration LNS samples, respectively, $A_{W,T_{cal}}$ is the absorbance of measurement seawater samples calculated by the established LNS model. In this way, the sea salt absorbance $A_{SS}$ caused by temperature change can be calculated as
(6)$$A_{SS}=E_{W,T_{m}}\times S,$$
where S represents the salinity of seawater. Therefore, by subtracting the temperature dependent data from the measured data, the absorbance of nitrate and other temperature independent components can be obtained as
(7)$$A\prime =A_{Meas}-A_{SS},$$
where $A_{Meas}$ is the absorbance of the measurement of seawater. Chromophoric dissolved organic matter (CDOM) in seawater exhibits a broadband absorption between 150 nm and 400 nm. Therefore, the estimation of the spectral behavior of CDOM needs to be subtracted from $A_{Meas}$ as CDOM is also a strong UV absorber. Three approaches have been tested to estimate the absorption of CDOM including linear functions [37,45], quadratic functions [33], and exponential functions [46]. However, the use of an exponential function was found to be problematic because the results obtained by the algorithm may be divergent if the fitting parameters are not close to the proper values. To simplify the model, the linear function for CDOM is established by the spectra data from 240 nm to 260 nm. Then the nitrate absorbance can be calculated as
(8)A″=A′−(e+fλ),
where e and f are the regression parameters for CDOM. Finally, based on the regression algorithm, the nitrate concentration calculation model can be applied.

#### 2.3.2. Weighted Average Kernel Partial Least Squares Algorithm

The kernel partial least squares (KPLS) algorithm integrates the advantages of principal component analysis (PCA), canonical correlation analysis (CCA), and linear regression analysis. It meets the conditions of $K(x_{i},x_{j})=\langle \varphi (x_{i}),\varphi (x_{j})\rangle ,x_{i},x_{j}\in R^{N}$. φ represents the nonlinear mapping, $x_{i}$ and $x_{j}$ are the data in data space $R^{N}$. KPLS algorithm cannot only deal with nonlinear relationships between variables, but also make full use of the spatial distribution information of samples to establish the model, which greatly improves the regression and prediction accuracy.

However, in the process of modeling, over-extraction of training set information will cause over-fitting. This makes the calculation model perform well on training set but has a large error on the testing set [47]. In order to solve this problem, Zhang and his colleagues proposed a multiple regression method based on a weighted average algorithm, with the results obtained relatively robust to over-fitting [48]. Based on this, an optimized weighted average kernel partial least squares (WA-KPLS) algorithm is proposed and established in this research, which is shown in Figure 3. Through the establishment and accuracy calculation of each sub model, the weight parameters will be obtained to effectively overcome the over-fitting problem and get more accurate prediction results.

Suppose that we have the independent variable matrix X with a size of n×p and dependent variable Y with size n×m, where n is the sample number, p and m are the dimensions of independent variable and dependent variable. Random sampling method is used here to randomly select η% samples of the training set for establishing the sub model with KPLS algorithm, which can be performed as follows [49]:
Randomly initialize u;$t=\varphi \varphi^{T}u=Ku,\frac{t}{\left\|t\right\|}\to t,K_{ij}=K(x_{i},x_{j})$$c=Y^{T}t$$u=Yc,\frac{u}{\left\|u\right\|}\to u$Repeat (2) to (4) steps until convergence;Calculate residual matrices of X and Y, $(I-tt^{T})K(I-tt^{T})\to K,Y-tt^{T}Y\to Y$;
where u and t are the principal components. Repeat the above steps until the result meets the requirement, and the regression coefficient of the model can be calculated as
(9)$$\beta =\varphi^{T}U{\left(T^{T}KU\right)}^{-1}T^{T}Y.$$

Therefore, the calculation model is established, and the regression data can be expressed as
(10)$$\hat{Y}=\varphi \beta =\varphi \varphi^{T}U{\left(T^{T}KU\right)}^{-1}T^{T}Y=KU{\left(T^{T}KU\right)}^{-1}T^{T}Y.$$

Based on the sub model, the prediction result for the training set is given by
(11)$${\hat{Y}}_{t}=\varphi_{t}\beta =K_{t}U{\left(T^{T}KU\right)}^{-1}T^{T}Y,$$
where $K_{t}=(I-\frac{1}{n}I_{t}I_{n}^{T})K(I-\frac{1}{n}I_{t}I_{n}^{T})$, $I_{t}$ is the vector whose elements are all 1 and dimension is the number of training set samples n, and $I_{n}$ is the vector whose elements are all 1 with dimension of n×1. D sub-models in total are established based on the training set. The results from each of these sub-models deviate differently from the reference concentrations. A weighted average of the sub-model results now forms the final result, with the weights being the inverse of the root mean square errors of the prediction (RMSEP). The RMSEP of $j_{th}$ sub model can be expressed as
(12)$$RMSEP_{j}=\sqrt{\frac{1}{n}\sum_{i=1}^{n}{\left(y_{i}-y_{i,p}^{j}\right)}^{2}},$$
where $y_{i}$ and $y_{i,p}^{j}$ are the reference concentration and predicted concentration of training set samples by $j_{th}$ sub-model. The smaller the $RMSEP_{j}$ is, the better performance the sub-model has, and the larger the weight should be. Therefore, the $j_{th}$ sub model weight $w_{j}$ can be defined as
(13)$$w_{j}=\frac{\frac{1}{RMSEP_{j}}}{\sum_{k=1}^{D}\frac{1}{RMSEP_{k}}},$$

The final predicted value $Y_{p}$ can be expressed as
(14)$$Y_{P}=\sum_{k=1}^{D}w_{k}Y_{kp},$$
where $Y_{kp}$ is the predicted value of the $k_{th}$ sub-model. In the process of modeling, each sub-model is considered as a basic learner. The proposed method is designed to adjust the proportion of different sub-model results in the final result by measuring the accuracy of each learner, so as to overcome the problem of over fitting [50].

### 2.4. Measurement Method

(1)A number of different artificial samples were created as follows to test the system sensitivity to different ions and to nitrate in seawater. To a basis of deionized water sodium chloride (0.55 mol/L), magnesium sulfate (28 mmol/L), sodium bicarbonate (2.3 mmol/L), sodium bromide (0.8 mmol/L), sodium nitrate (30 µmol/L), sodium humate (5 µmol/L), sodium dihydrogen phosphate (2.26 µmol/L), and sodium nitrite (0.22 µmol/L) were added [51]. The seawater from Western Pacific and Aoshan Bay, Qingdao of China were filtered using a 0.45 μm filter membrane. AutoAnalyzer 3 (SEAL, Germany) is used to measure the background nitrate concentration in Western Pacific and Aoshan Bay seawater. The background nitrate concentration in Western Pacific seawater is lower than 0.1 µmol/L. The background nitrate concentration in Aoshan Bay seawater is 1.59 µmol/L. Different concentrations of nitrate (0–100 µmol/L) in seawater were created with a basis of LNS and Aoshan Bay seawater. The LNS sample, Western Pacific seawater samples, and Aoshan Bay seawater samples were frozen at −20 °C in clean high-density polyethylene bottles. These seawater samples with different nitrate concentrations were finished measuring within 5 days.(2)Based on the designed system, the absorbance of the different solutions and of seawater samples was measured at different temperatures (4–25 °C at 1 °C intervals) to establish the temperature dependency of the measurements. All measurements were done after the temperature had stabilized. The results from these measurements were used to select the optimal wavelength range and establish the temperature and salinity correction model by the measurement data.(3)The LNS samples, Western Pacific seawater samples, and Aoshan Bay seawater samples with addition of nitrate concentrations were measured by the system. The nitrate calculation models based on different regression algorithms were established.(4)The nitrate concentrations in Western Pacific seawater samples and Aoshan Bay seawater samples were calculated and analyzed by using different regression algorithms.

## 3. Results

### 3.1. Influence of Ions and Temperature

The main components in seawater were measured and analyzed to determine the main interfering substances and optimal modeling wavelength range, and the results are shown in Figure 4. It can be seen that in the range between 200 nm and 240 nm, bromide ions and chloride ions are the main factors that have a great influence on the seawater absorbance data, while the absorbance of bicarbonate, sulfate, and other particles are small. As the absorbance of chloride ion is high for wavelength lower than 208 nm, which could submerge the absorbance signal of nitrate, and no obvious absorbance for wavelength longer than 240 nm, spectral data from 208 nm to 240 nm are selected for data modeling and calculation in order to reduce the influence of system noise and absorbance of other components on the calculation model.

The temperature dependence of pure components and LNS is illustrated in Figure 5. It can be seen that LNS and bromide absorbances increase at higher temperature, and the temperature slope of LNS data is almost similar to the slope generated by NaBr solutions with the same bromide concentrations, whereas the absorbance of other pure components, including nitrate, do not show a temperature dependence. The LNS absorbance model with temperature influence is shown in Figure 6, where x-axis represents the normalized wavelength, y-axis represents the normalized temperature, z-axis represents the absorption of LNS, and the color scale represents different absorbance. The *R*^2^ is 0.9971, RMSE is of 0.02, and the sum of squares due to error (SSE) is of 1.508 of the LNS model.

### 3.2. A Nitrate Calculation Model Based on the WA-KPLS Algorithm with Temperature and Salinity Correction

A training set with 614 samples was prepared with different nitrate concentrations (0–100 µmol/L) to establish the model. The temperatures ranged from 4 °C to 25 °C. After collecting the measurement data, WA-KPLS algorithm with temperature and salinity correction (TSC-WA-KPLS) is carried out to establish the calculation model. The optimal number of principal components for the TSC-WA-KPLS method can be assessed by the determination coefficient *R*^2^ and the RMSEP. The equations are:(15)$$R^{2}=1-\frac{\sum_{i=1}^{n}{\left(y_{i}-y_{i,p}\right)}^{2}}{\sum_{i=1}^{n}{\left(y_{i}-\hat{y}\right)}^{2}},$$
(16)$$RMSEP=\sqrt{\frac{1}{n}\sum_{i=1}^{n}{\left(y_{i}-y_{i,p}\right)}^{2}},$$
where $y_{i}$ and $y_{i,p}$ are the reference and predicted concentrations of the samples, respectively. $\hat{y}$ is the mean value of reference concentration and n is the sample number. The smaller RMSEP is, the higher the accuracy is. When *R*^2^ is larger than 0.9995, it is considered that the model has reached the requirement and then the number of principal components can be determined. The results of *R*^2^ and RMSEP are shown in Figure 7. It can be seen from the figure that when the number of principal component reaches 12, RMSEP of the model is less than 1 µmol/L and *R*^2^ is 0.9995, which means that the established model meets our requirement. Therefore, in this study, the number of TSC-WA-KPLS principal components is selected as 12.

### 3.3. Prediction Results for Aoshan Bay Seawater and Western Pacific Seawater

In order to verify the reliability and accuracy of the established models, 110 Aoshan Bay seawater and 94 Western Pacific seawater samples with nitrate concentrations ranging from 0 μmol/L to 100 μmol/L were measured with the system. Meanwhile, the prediction results are also compared with MLR and ISUS algorithms, and the results are shown in Figure 8 and Figure 9, where (a,b) represent the residual results of TSC-WA-KPLS and TSC-MLR algorithms, (d,e) represent the frequency count results of TSC-WA-KPLS and TSC-MLR algorithms, (g) represents the linear correction with the reference concentrations for ISUS algorithm, (c,f) represent the residual and the frequency count results of ISUS algorithm after linear correction. Sample 1 to sample 60 in Figure 8 are the Aoshan Bay seawater samples with nitrate concentrations from 1 μmol/L to 10 μmol/L, and the rest are the Aoshan Bay seawater samples with nitrate concentrations from 10 μmol/L to 100 μmol/L. Sample 1 to sample 52 in Figure 9 are the Western Pacific seawater samples with nitrate concentrations from 1 μmol/L to 10 μmol/L, and the rest are the Western Pacific seawater samples with nitrate concentrations from 10 μmol/L to 100 μmol/L. It can be seen from Figure 8a–c that the samples with high nitrate concentrations (samples 61 to 100) have a larger prediction error than those with low nitrate concentrations (samples 1 to 60). From Figure 8d–f we can see that the residuals of TSC-WA-KPLS and TSC-MLR algorithm are obviously more concentrated in the range between −2 μmol/L and 2 μmol/L than those of ISUS algorithm. Similar results are found for the Western Pacific seawater samples in Figure 9a–f, except that the residuals of TSC-WA-KPLS and ISUS algorithm are obviously more concentrated in the range between −2 μmol/L and 2 μmol/L than those of TSC-MLR algorithm.

Table 2 shows the prediction results based on different algorithms, including the existing algorithms of MLR and ISUS. For the samples with high nitrate concentrations (10 μmol/L–100 μmol/L), TSC-WA-KPLS has the smallest relative error range. For the Aoshan Bay seawater, TSC-WA-KPLS has the smallest error range (±2 μmol/L) and smallest RMSEP (0.67 μmol/L) among three algorithms. For the Western Pacific seawater, the error ranges of TSC-WA-KPLS and ISUS are smaller than those of TSC-MLR, which are both between ±3.3 μmol/L. However, the *R*^2^ of the TSC-WA-KPLS is larger, which indicates that the calculation model established by TSC-WA-KPLS has better prediction performance. Therefore, TSC-WA-KPLS has the best prediction performance with *R*^2^ of 0.9996 and 0.9987 for the Aoshan Bay seawater and Western Pacific seawater, respectively.

## 4. Discussion

A different method for measuring nitrate concentration in seawater based on deep-ultraviolet spectral sensor is proposed and studied in this paper. When using ultraviolet spectrometer to determine the nitrate concentration in seawater interference by other substances in the seawater, such as bromide, chloride, organic matter, nitrite, sulfate, bicarbonate, etc. [52], occurs. If all these substances were considered as interference factors, the calculation of nitrate would be very complex. On the premise of keeping the accuracy of nitrate measurement, the absorbance characteristics of several compositions in seawater in the deep-UV range were measured in the wavelength range of 208 nm–240 nm in this study, which could minimize the impact of background noise and improve the nitrate calculation model [37]. 

In order to subtract the temperature influence from the measured absorbance data for a more accurate nitrate calculation model, the relationship between different ions and temperature was measured and analyzed in this study. The results showed that the bromide absorbance is dependent on temperature, with the same trend as that of LNS, which was consistent with the results in previous literature [52,53]. This phenomenon is due to the fact that the absorbance of bromide in deep UV was generated by charge transfer to the solvent complex, and the interaction between bromide and solvent would produce strong temperature dependence, resulting in the bromide absorbance increase with the temperature increase [53]. However, as the absorbance of nitrate was caused by the π→π* transition within the molecule, there was no obvious temperature dependence [54]. This result also verified that the changes in sea salt absorbance with temperature were primarily due to the change of bromide absorbance. Laboratory studies by several research groups also showed a strong dependence between seawater absorbance and temperature [37,55], as the seawater absorbance was mainly determined by the ultraviolet absorbance of bromide [8,44]. Since bromide is stable with salinity, the bromide concentration is reported as a salinity [56]. Therefore, by subtracting the absorbance of the sea salt caused by the temperature change, the influence of bromide can be removed [37,45]. Furthermore, the simple linear fitting model is used here to decrease the absorbance effect of CDOM and particles offset. However, this is the rough estimation of the behavior of CDOM in the UV range as the CDOM concentration may vary with time and space [57].

By analyzing the absorbance characteristics of various components at different temperatures, the TSC algorithm based on low nutrient seawater was established within the temperature range of 4–25 °C. Through the application of TSC algorithm and regression algorithms in deep-UV spectral analysis, different nitrate prediction models were established. The calculation results of Aoshan Bay seawater and Western Pacific seawater were compared and analyzed by different models based on TSC-WA-KPLS was established. Moreover, TSC-MLR and ISUS algorithm are also applied on the seawater data to make the comparison with the proposed algorithms. The experimental results showed that: 

For the Aoshan Bay seawater and Western Pacific seawater samples, the model based on TSC-WA-KPLS has the best prediction performance, with the largest coefficient *R*^2^ and lowest RMSEP. The algorithms based on TSC-WA-KPLS can predict the samples well, and the residuals of low nitrate concentration samples are smaller than those of high nitrate concentration samples. Furthermore, the prediction ability of TSC-WA-KPLS algorithm is better than that of TSC-MLR algorithm, which shows that the compensation effect and generalization ability of TSC-WA-KPLS algorithm are better. MLR algorithm is simple, but the calculation performance will not be as good as nonlinear multiple regression model for the data with nonlinear relation. Since we only measure the seawater samples in the laboratory, the influence of depth has not been involved. The temperature and salinity dependency of the TSC-WA-KPLS residuals are calculated and the results are shown in Figure 10, where Figure 10a,b are the results of Aoshan Bay seawater, Figure 10c,d are the results of Western Pacific seawater. It can be seen from Figure 10a,c that the residuals of the samples do not show any temperature dependency. The salinity ranges of the Aoshan Bay seawater samples and Western Pacific seawater samples are very small in Figure 10b,d, and the residuals at the same temperature also has no obvious salinity dependency. Therefore, the TSC-WA-KPLS algorithm is more suitable for practical measurement.

The measurement errors for Aoshan Bay seawater and Western Pacific seawater might come from several reasons. Firstly, the sample aging issue may be introduced due to lab analysis of the seawater samples, which was performed on different days. Secondly, the presented method is based on the system using deuterium lamp with broadband light, which does not comply with the monochromatic light incident requirement of Lambert Beer’s law. In this case, the broadband light may excite other components in seawater to produce optical signals that might result in measurement error. Deep ultraviolet LEDs could be considered as the light source in the future because near mono-chromaticity can be provided. Thirdly, bending of the optical fiber in the operation process might cause variation of absorbance, which will introduce measurement error. Therefore, all optical fibers are tried to keep straight and fixed in order to avoid this error. 

To analyze the ability of TSC-WA-KPLS to handle data that are not in the training range, we set the training seawater samples with concentration range of 4 µmol/L–80 µmol/L, 6 µmol/L–80 µmol/L, 8 µmol/L–80 µmol/L, and 10 µmol/L–80 µmol/L, respectively, and predict the seawater samples with nitrate concentration outside of the known range. The measurement results are shown in Figure 11, where a and b illustrate the results for the prediction samples with nitrate concentration of 0 µmol/L–3 µmol/L and 90 µmol/L–100 µmol/L when the training seawater samples are with concentration range between 4 µmol/L and 80 µmol/L. For the samples with low nitrate concentration from 0 µmol/L to 3 µmol/L, the number of the samples with error range within ±2 µmol/L accounted for 100%. For the samples with high nitrate concentration from 80 µmol/L to 100 µmol/L, the number of the samples with error range within ±2 µmol/L accounted for 0%. Figure 11c,d shows the results for the prediction samples with nitrate concentration of 0 µmol/L–5 µmol/L and 90 µmol/L–100 µmol/L when the training seawater samples are with concentration range between 6 µmol/L and 80 µmol/L. For the samples with low nitrate concentration from 0 µmol/L to 5 µmol/L, the number of the samples with error range within ±2 µmol/L accounted for 92.70%. For the samples with high nitrate concentration from 90 µmol/L to 100 µmol/L, the number of the samples with error range within ±2 µmol/L accounted for 0%. Figure 11e,f shows the results for the prediction samples with nitrate concentration of 0 µmol/L–7 µmol/L and 90 µmol/L–100 µmol/L when the training seawater samples are with concentration range between 8 µmol/L and 80 µmol/L. For the samples with low nitrate concentration from 0 µmol/L to 7 µmol/L, the number of the samples with error range within ±2 µmol/L accounted for 88.48%. For the samples with high nitrate concentration from 90 µmol/L to 100 µmol/L, the number of the samples with error range within ±2 µmol/L accounted for 0%. Figure 11g,h shows the results for the prediction samples with nitrate concentration of 0 µmol/L–9 µmol/L and 90 µmol/L–100 µmol/L when the training seawater samples are with concentration range between 10 µmol/L and 80 µmol/L. For the samples with low nitrate concentration from 0 µmol/L to 9 µmol/L, the number of the samples with error range within ±2 µmol/L accounted for 28.01%. For the samples with high nitrate concentration from 90 µmol/L to 100 µmol/L, the number of the samples with error range within ±2 µmol/L still accounted for 0%. Therefore, the prediction sample, with concentration less than 8 µmol/L out of the training range, could get relatively accurate result when the measurement accuracy is evaluated by residuals. When the concentration of the prediction sample is more than 10 µmol/L out of the training range, the prediction accuracy of the model will be reduced, and the prediction result cannot be trusted.

In the algorithm of ISUS, the calculation model is established with the wavelength range from 217 nm and 240 nm. The measurement resolution is approximately 1 nm [37]. In this study, a UV spectrometer with higher resolution (0.4 nm) is used in the measurement system, and the wavelength range of the calculation model with TSC algorithm is from 208 nm to 240 nm. With the higher resolution and wider wavelength range, more measurement points can be achieved. However, the data may be more susceptible to noise with smaller pixels and thus cause measurement error. The absorbance of the LNS ($A_{LNS}$) was measured by the system and calculated by the Equation (4). After that, $A_{LNS}$ was compared with the oligotrophic seawater (in oligotrophic regions of Pacific Ocean) absorbance fitted by ISUS algorithm in reference [37]. The fitting function of the oligotrophic seawater absorbance by ISUS algorithm is expressed as: (17)$$A_{OS}=(A+B\times T)\times \exp((C+D\times T)\times W),$$
where T is the sample temperature in degC, W is the wavelength minus 210 nm. The resultant regression parameters A, B, C, and D are 1.1500276, 0.02840, −0.3101349, and 0.001222, respectively. The result is shown in Figure 12, where the horizontal-axis represents the wavelength and the vertical-axis represents the absorbance difference (subtracting ISUS oligotrophic seawater absorbance from the measured Western Pacific seawater absorbance). As the wavelengths collected by the spectrometer in this system is not completely consistent with those of ISUS, the nearest wavelengths are selected for calculation. It can be seen in Figure 12 that the Western Pacific seawater absorbance measured by this system is smaller than the oligotrophic seawater absorbance measured by ISUS system. The absorbance models of the Pacific seawater calculated by the presented system and the ISUS system are not exactly the same. When using the measurement data of the system to calculate the nitrate concentration by the ISUS algorithm, there are still large errors with the reference concentrations. Therefore, these results need to be linear corrected again to the final nitrate concentration value (shown in Figure 8g and Figure 9g).

Though previous methods allow to get quite good accuracy of nitrate in seawater [37,52], the results demonstrated in this study with a different algorithm could measure nitrate concentration in Aoshan Bay seawater and Western Pacific seawater with higher accuracy. KPLS is an important method in multiple regression analysis, which can accurately calculate the correlation between various factors and the degree of regression fitting. Therefore, based on KPLS algorithm, the spectral data (independent variables) can be mapped into a high-dimensional feature space by the kernel function, then the PLS regression model is established to calculate the results. Compared with existing methods (including ISUS algorithm and MLR algorithm), the proposed method can more accurately predict the nitrate concentration in a specific region. However, the TSC-WA-KPLS algorithm does not have the ability of self-learning. When the sample concentration is more than 10 µmol/L out of the training range, the proposed method cannot get accurate results, as the results shown in Figure 11. The error of the proposed TSC-WA-KPLS algorithm is ±3.2 µmol/L for the Western Pacific seawater and ±2 µmol/L for the Aoshan Bay seawater. Organic matter and turbidity in seawater are also the influence factors in nitrate measurement. Organic matter can be divided into dissolved organic matter and particulate organic matter. In this paper, all the seawater samples were filtered using 0.45 µm filter membrane (mentioned in Section 2.4), so the influence of turbidity and particulate organic matter on nitrate measurement can be removed. However, for the in-situ measurement, this method needs to be improved as turbidity correction algorithm is not included. Therefore, turbidity correction algorithm will be studied for the actual unfiltered seawater in our future research.

## Figures and Tables

**Figure 1 sensors-21-00965-f001:**
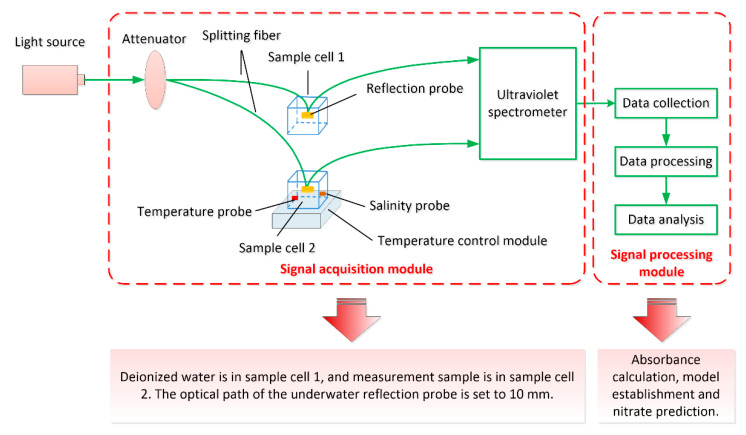
Structure of the measurement system.

**Figure 2 sensors-21-00965-f002:**
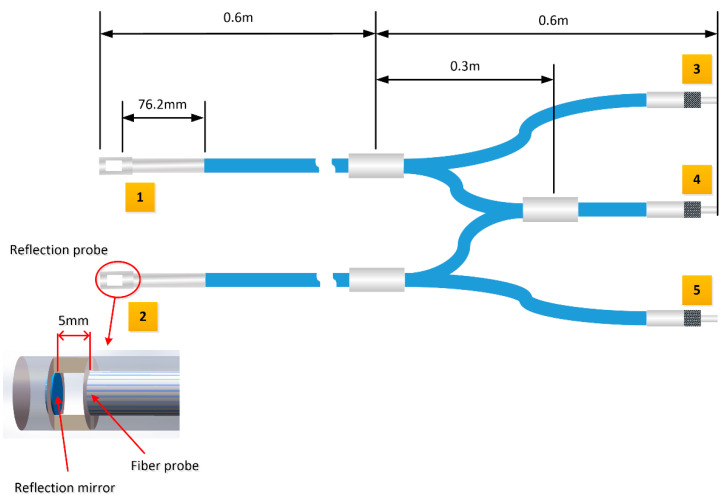
Structure of the UV fiber splitter.

**Figure 3 sensors-21-00965-f003:**
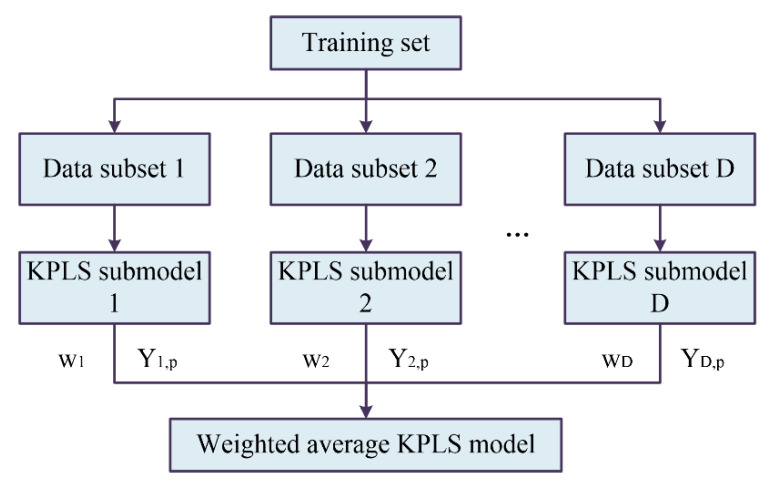
Establishment of the weighted average kernel partial least squares (KPLS) model.

**Figure 4 sensors-21-00965-f004:**
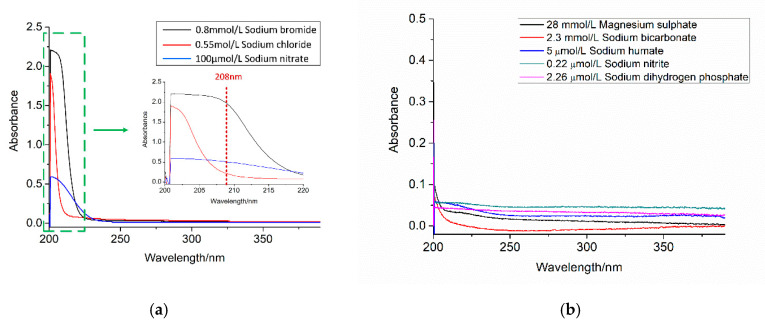
Absorbance of pure solutions. (**a**) NaCl, NaBr, and NaNO_3_; (**b**) MgSO_4_, NaHCO_3_, sodium humate, NaH_2_PO_4_, and NaNO_2_.

**Figure 5 sensors-21-00965-f005:**
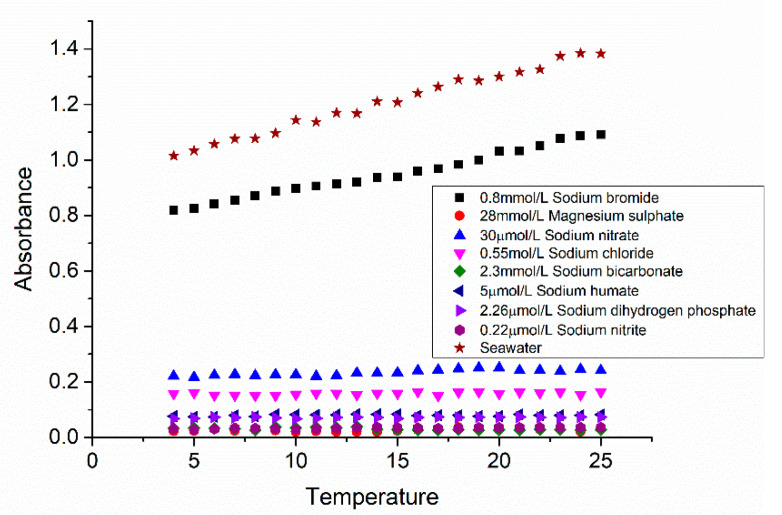
Relationship between absorbance and temperature of pure solutions and seawater at 210.029 nm.

**Figure 6 sensors-21-00965-f006:**
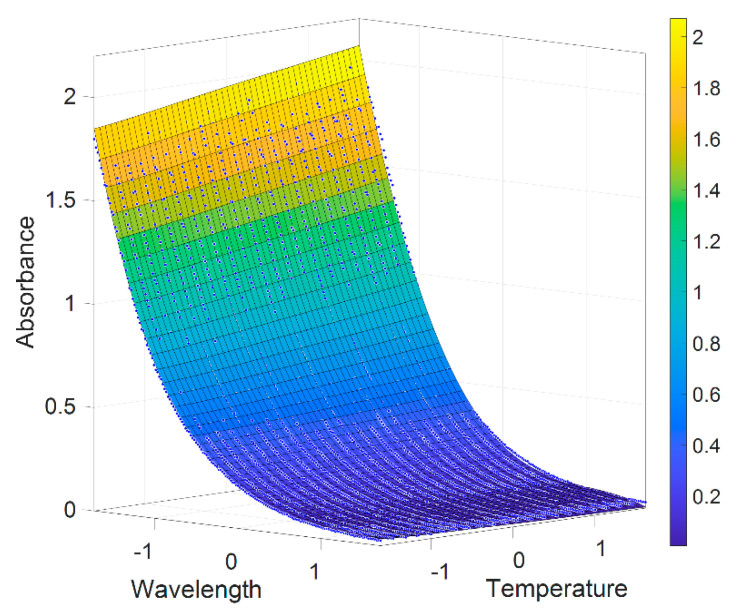
Three-dimensional absorption model of low nutrient seawater (LNS). In order to obtain better LNS fitting model, temperature and wavelength are normalized by the equation $\frac{t-mean(t)}{std(t)}$ and $\frac{wl-mean(wl)}{std(wl)}$.

**Figure 7 sensors-21-00965-f007:**
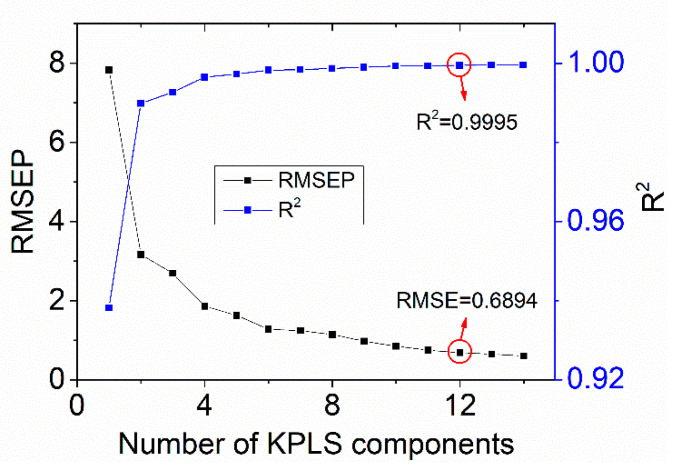
Relationship between principal component number and root mean square errors of the prediction (RMSEP), *R*^2^.

**Figure 8 sensors-21-00965-f008:**
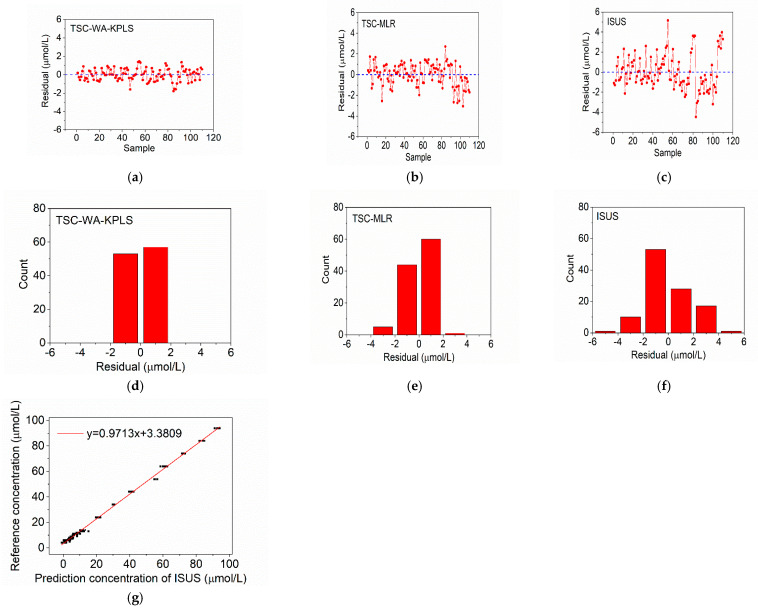
Prediction results of seawater samples from Aoshan Bay. (**a**) Residual of temperature and salinity correction weighted average kernel partial least squares (TSC-WA-KPLS) algorithm; (**b**) residual of TSC-multiple linear regression (MLR) algorithm; (**c**) residual of ISUS algorithm after linear calculation; (**d**) frequency count of TSC-WA-KPLS; (**e**) frequency count of TSC-MLR; (**f**) frequency count of ISUS model after linear calculation; (**g**) linear correction for ISUS algorithm.

**Figure 9 sensors-21-00965-f009:**
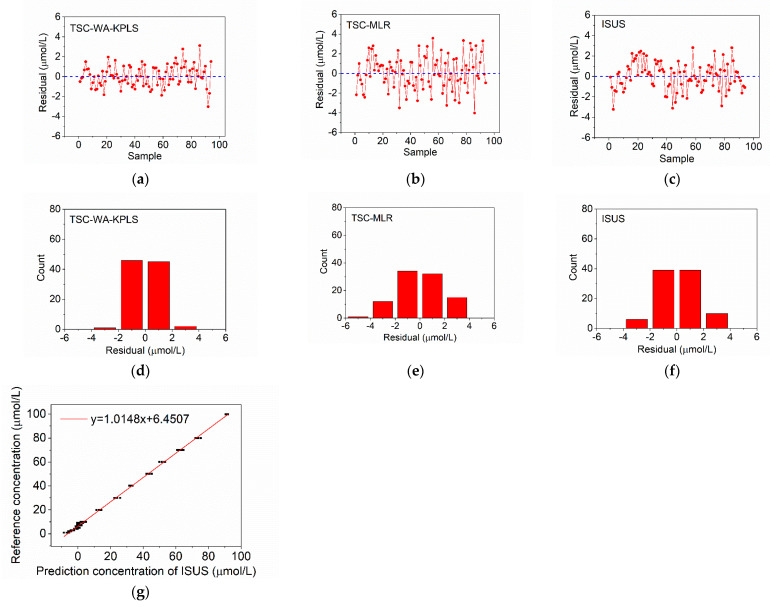
Prediction results of seawater samples from Western Pacific. (**a**) Residual of TSC-WA-KPLS algorithm; (**b**) residual of TSC-MLR algorithm; (**c**) residual of ISUS algorithm after linear calculation; (**d**) frequency count of TSC-WA-KPLS; (**e**) frequency count of TSC-MLR; (**f**) frequency count of ISUS model after linear calculation; (**g**) linear correction for ISUS algorithm.

**Figure 10 sensors-21-00965-f010:**
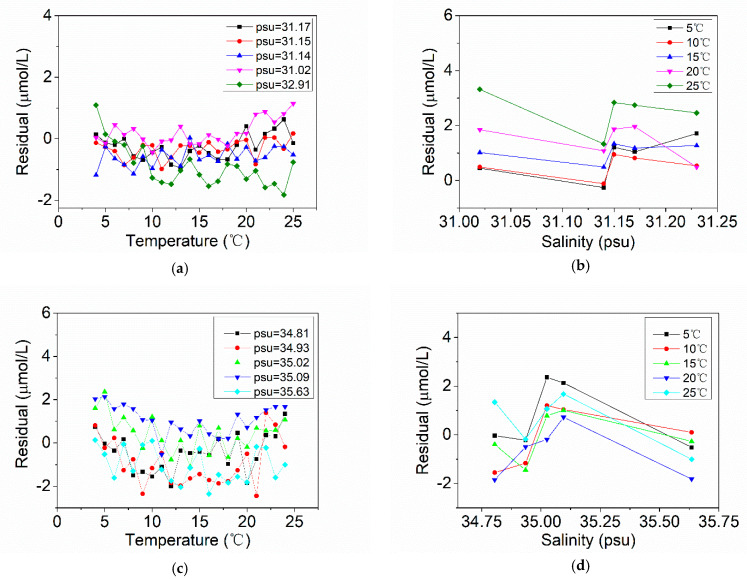
Temperature and salinity dependency results. (**a**) Temperature dependency of Aoshan Bay seawater samples; (**b**) salinity dependency of Aoshan Bay seawater samples; (**c**) temperature dependency of Western Pacific seawater samples; (**d**) salinity dependency of Western Pacific seawater samples.

**Figure 11 sensors-21-00965-f011:**
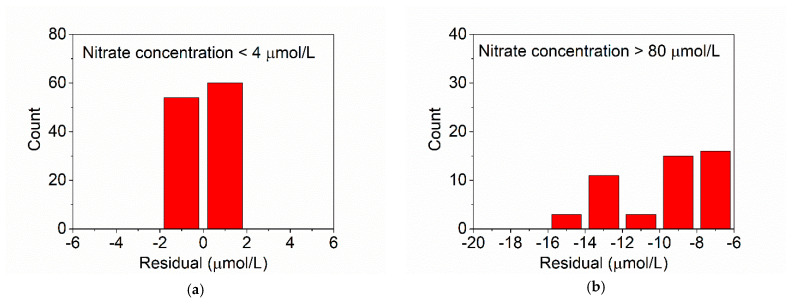
Prediction results for seawater samples with nitrate concentration outside of the known range. (**a**) Prediction samples with nitrate concentration of 0 µmol/L–3 µmol/L with training samples with concentration of 4 µmol/L–80 µmol/L; (**b**) prediction samples with nitrate concentration of 90 µmol/L–100 µmol/L with training samples with concentration of 4 µmol/L–80 µmol/L; (**c**) prediction samples with nitrate concentration of 0 µmol/L–5 µmol/L with training samples with concentration of 6 µmol/L–80 µmol/L; (**d**) prediction samples with nitrate concentration of 90 µmol/L–100 µmol/L with training samples with concentration of 6 µmol/L–80 µmol/L; (**e**) prediction samples with nitrate concentration of 0 µmol/L–7 µmol/L with training samples with concentration of 8 µmol/L–80 µmol/L; (**f**) prediction samples with nitrate concentration of 90 µmol/L–100 µmol/L with training samples with concentration of 8 µmol/L–80 µmol/L; (**g**) prediction samples with nitrate concentration of 0 µmol/L–9 µmol/L with training samples with concentration of 10 µmol/L–80 µmol/L; (**h**) prediction samples with nitrate concentration of 90 µmol/L–100 µmol/L with training samples with concentration of 10 µmol/L–80 µmol/L.

**Figure 12 sensors-21-00965-f012:**
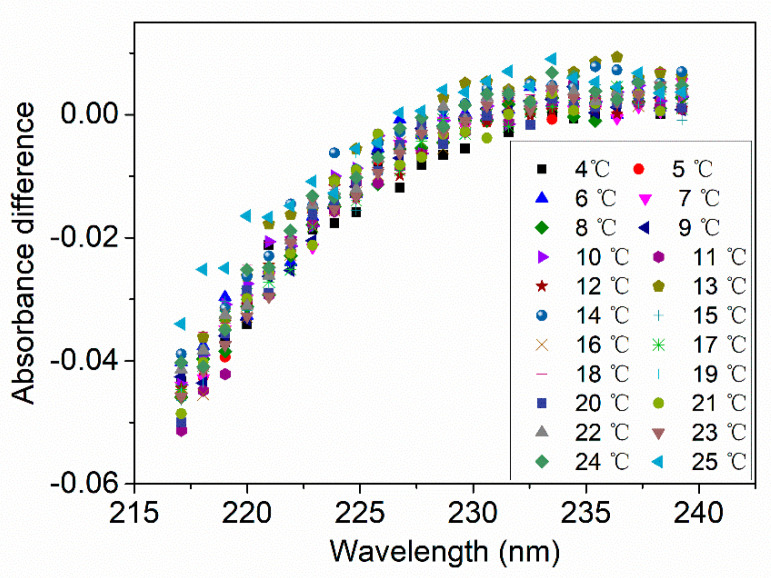
Absorbance difference ($A_{LNS}-A_{OS}$) between the LNS in this paper and the oligotrophic seawater in reference [37]. $A_{LNS}$ can be calculated by Equation (4) and $A_{OS}$ can be calculated by Equation (17).

**Table 1 sensors-21-00965-t001:** Specifications of the system.

Component	Parameter	Description
Light source	Type	Deuterium lamp (DH-2000-DUV)
	Power consumption	585 µW
	Lifetime	1000 h
	Output wavelength	190 nm–400 nm
	Output stability	Less than 5 × 10^−6^ peak to peak (0.1–10.0 Hz)
	Output drift	Less than 0.01% per hour
UV spectrometer	Wavelength range	200 nm–385 nm
	Optical resolution	0.4 nm
	Entrance slit	10 µm
	Grating	GRATING_#H48
	Dark noise	2.5 counts RMS
	Signal to noise ratio	1000:1 (single acquisition)
	Thermal stability	0.01 pixels/°C

**Table 2 sensors-21-00965-t002:** Prediction results of different algorithms.

Sample	Algorithm Model	RMSEP (µmol/L)	Error Range (µmol/L)	*R* ^2^
Aoshan Bay seawater	TSC-WA-KPLS	0.67	[−1.86, 1.53]	0.9996
	TSC-MLR	1.10	[−3.06, 2.72]	0.9989
	ISUS	1.75	[−4.47, 5.16]	0.9971
Western Pacific seawater	TSC-WA-KPLS	1.08	[−3.00, 3.17]	0.9987
	TSC-MLR	1.72	[−4.01, 3.58]	0.9967
	ISUS	1.36	[−3.22, 2.81]	0.9980

## Data Availability

Not applicable.

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
