# Peer review of "An Improved Algorithm for Measuring Nitrate Concentrations in Seawater Based on Deep-Ultraviolet Spectrophotometry: A Case Study of the Aoshan Bay Seawater and Western Pacific Seawater"

_sensors, 2021, doi:10.3390/s21030965_

Round 1

Reviewer 1 Report

The accurate determination of nitrate in seawater based on UV sensors is facing the challenges from complex sample matrix, such as interferences from organic matters and bromide. The authors developed a detection system and TSC-WA-KPLS algorithm to improve the accuracy and applicability of the UV-based nitrate analysis. Spiked samples in two different seawater matrices (e.g. open ocean and coastal sea) were measured. In all, the manuscript could be accepted after minor revision. Some detailed questions and comments are shown below: 1. Did the authors evaluate the interference from organic matters? It may not be a problem for open ocean samples, but for estuarine and coastal samples, it is needed to consider this interference. 2. It is recommended to state the information of low nutrient seawater (LNS) when this abbreviation firstly showed in the manuscript. 3. The reference for recipe of the artificial samples should be added. 4. Pay attention to the significant figures of concentration, e.g. 1.587 μmol/L might be 1.59 μmol/L, because AA3 can not accurately measure 1 nmol/L nitrate. Also, in the abstract, 1.0782 μmol/L has too many significant figures. 5. Figure 11, use “μmol/L” not “umol/L”.

Reviewer 2 Report

General

The paper is well constructed and written. The English is at a professional level but with a few errors which don’t change the meaning.

Title

“A different approach” implies (to me) a different physical setup + the improved algorithm. In the paper the setup is standard with an improved algorithm added. Consider modifying the title to emphasise the algorithm.

Introduction

Line 55: How does reference (20) support the claim that systems using chemical reagents are not suitable for in-situ measurements.

Some systems e.g. https://aquamonitrix.com/ can operate in-situ. (although nowhere near as robust as an optical probe.)

Line 105: “improve the quality of the concentrations” implies that the water quality would be improved. This should be re-phrased.

Materials and methods

Please provide details of the reflectance probe.

Have you considered using an LED as a light source? These exist with wavelengths down to 100 nm and can provide near mono-chromaticity.

In Table 1 it is stated

Deuterium lamp (DH-2000) has Output wavelength 190 nm - 400 nm.

I checked the OceanOptics site and it seems the DH-2000-DUV is capable of this, not the basic DH-2000.

Line 196: PSU is not define in the paper.  Consider adding the definition ‘Practical Salinity Unit’

Line 291 to 297: Please provide further details of the storage conditions of the LNS seawater. (i.e. temperature and duration). Nutrient concentrations in natural waters change over time so samples are typically frozen or used within 48 hours.

Line 304 to 306: This sentence construction doesn’t match with the other numbered steps under Measurement method.

Discussion

Line 413: Reference (52) is about Freshwater. Does this support the claim about the interferences in Seawater?

Line 436: Is “conservative ” the correct word? Your meaning is clear but I haven’t seen the word used like this before.

Line 435: Bromide is mentioned

Line 438: Bromine is mentioned.

Is this intentional? This occurs a few times in the document.

Line 475: “lab analysis of the sea water samples which were performed on different days.” This could indicate a sample aging issue due to the storage conditions.

Line 477: “does not comply with the monochromatic light incident requirement of Lambert Beer’s law.” Consider using an LED.

Line 480 to 484: this sentence does not read well. Consider re-phrasing.

Line 494: What is the basis of the ± 2 μmol/L error range? Is that a standard?

Line 539: “was measured using the measurement system”. Rephrase this

Line 540: “oligotrophic seawater fitted in ISUS algorithm.” Is this process detailed in the paper?

Figure 11: Presentation of residual data

Would it be better to represent the error of the measurement as a percentage of the actual value?

e.g. if the concentration is 4 umol and the error is +-2 umol then the error is 50% whereas if the concentration is 50 umol and the error is 2 umol then the percentage error is 4%

Line 536: “In this study, a UV spectrometer with higher resolution (0.4 nm) is used in the measurement system,” How does that make a difference to the results?

Line541: Figure 13?

Figure 12: The caption gives little information about the graph.

Line 548: “Therefore, the linear deviation usually occurs between the calculation results and the reference concentrations.” The meaning of this sentence is unclear.

Line 534 to 550.

This whole section including Figure 12 is unclear to me. Did the Author use the ISUS instrument to analyse the Oligotrophic water? If not, then where does the data originate?

General notes

Turbidity

Chemical interferences and temperature are well dealt with in the paper but turbidity is not. The samples used are LNS spiked with Nitrate.

If the intended use is for in-situ sensors (stated on line 25 and line 40) then turbidity should be dealt with experimentally or at least discussed.

The TRIOS NICO probe measures turbidity at 360 nm in order to compensate the Nitrate measurement.
